# A Beacon in the Dark: Grey Literature Data Mining and Machine Learning Enlightening Historical Plankton Seasonality Dynamics in the Ligurian Sea

Alice Guzzi [1,2,*] , Stefano Schiaparelli [1,2,3] , Maria Balan [1] and Marco Grillo [3,4]

1   Department of Earth, Environmental and Life Sciences (DISTAV), University of Genoa, Corso Europa 26, 16132 Genoa, Italy; stefano.schiaparelli@unige.it (S.S.); mariabalan42@gmail.com (M.B.)
2   National Biodiversity Future Center (NBFC), Piazza Marina 61, 90100 Palermo, Italy
3   Italian National Antarctic Museum (MNA, Section of Genoa), Viale Benedetto XV No. 5, 16132 Genoa, Italy; grillomarco94@gmail.com
4   Department of Physical Sciences, Earth and Environment (DSFTA), University of Siena, 53100 Siena, Italy
*   Correspondence: aliceguzzi@libero.it

**Abstract:** The Mediterranean Sea, as one of the world's most climate-sensitive regions, faces significant environmental changes due to rising temperatures. Zooplankton communities, particularly copepods, play a vital role in marine ecosystems, yet their distribution dynamics remain poorly understood, especially in the Ligurian Sea. Leveraging open-source software and environmental data, this study adapted a methodology to model copepod distributions from 1985 to 1986 in the Portofino Promontory ecosystem using the Random Forest machine learning algorithm to produce the first abundance and distribution maps of the area. Five copepod genera were studied across different trophic guilds, revealing habitat preferences and ecological fluctuations throughout the seasons. The assessment of model accuracy through symmetric mean absolute percentage error (sMAPE) highlighted the variability in copepod dynamics influenced by environmental factors. While certain genera exhibited higher predictive accuracy during specific seasons, others posed challenges due to ecological complexities. This study underscores the importance of species-specific responses and environmental variability in predictive modeling. Moreover, this study represents the first attempt to model copepod distribution in the Ligurian Sea, shedding light on their ecological niches and historical spatial dynamics. The study adhered to FAIR principles, repurposing historical data to generate three-dimensional predictive maps, enhancing our understanding of copepod biodiversity. Future studies will focus on developing abundance distribution models using machine learning and artificial intelligence to predict copepod standing crop in the Ligurian Sea with greater precision. This integrated approach advances knowledge of copepod ecology in the Mediterranean and sets a precedent for integrating historical data with contemporary methodologies to elucidate marine ecosystem dynamics.

**Keywords:** machine learning; copepods; species distribution models (SDMs); Ligurian Sea; open source; FAIR data





## 1. Introduction

Climate change stands as an escalating and pervasive worldwide threat to biodiversity and ecosystems' functioning [1], posing an intricate intergovernmental challenge, across various domains, in terms of conservation, management and economical aspects [2]. Major drivers, such as ocean warming and acidification, among others, exert a profound influence on global biodiversity, causing significant alterations in the dynamics of marine communities [3–9]. Intrinsic characteristics of marine organisms, such as physiological tolerance [10–13], larval ecology [14,15], ecological plasticity and rapid adaptation to environmental variations [11,16–18], are expected to be affect in response to these drivers.

Consequential ecological effects are also expected to vary across temporal and spatial scales due to extrinsic factors, such as the rate of environmental change [19,20], the magnitude of environmental variability [21] and the differences in environmental conditions between regions and habitats [22].

The semi-enclosed marine region of Mediterranean Sea, the second largest marine biodiversity hotspot globally [23], is projected to be highly affected by the altering climate conditions at a faster pace than the global average [24]. Although this region represents merely 0.32% of the total ocean volume, its distinctive geomorphological history has resulted in an exceptionally rich biodiversity, encompassing 7–10% of all identified marine species, including a significant proportion of endemic taxa [23,25]. The Mediterranean basin has experienced a significant rise in sea surface temperature (SST) since the beginning of satellite records in 1982, exhibiting a warming trend in the period from 1982 to 2019 at a rate of 0.38 °C per decade [26]. This rate is over three times greater than the global average of 0.11 °C per decade [27]. This presents specific concern and attention to the Mediterranean Sea due to the numerous instances of heat waves [28] triggered by the constant temperature rise. Currently, the average temperature stands at $1.2 \pm 0.23$ °C (mean $\pm$ standard deviation of absolute annual anomalies) higher than the earliest available satellite data (corresponding to the years 1982–1986), indicating a substantial temperature increase [26]. This poses significant risks to the diverse marine life in the region, with notable impacts on biodiversity (e.g., [26,28–30]). To efficiently tackle any of these challenges, increasing the resolution of observational and monitoring capacities over relevant spatial, temporal and taxonomic scales is essential. Pelagic ecosystems are the first responder to changing conditions, exhibiting shifts in species distribution across trophic levels [31–36]. Environmental changes, such as temperature, salinity and chlorophyll concentration, impact marine life from plankton to mammals. In pelagic ecosystems, zooplankton, particularly copepods, play a crucial role in connecting primary producers to higher-level consumers in marine food webs [37]. The distribution of zooplankton is intricately regulated by various factors, encompassing physical and chemical constraints [38,39]. Additionally, biological interactions, including predator–prey dynamics, symbiosis, parasitism and commensalism, further influence their distribution [40]. Due to the short generation times and fecundity of zooplankton, their populations can rapidly respond (<1 year) to changes in the marine environment [41], of course including those of anthropogenic origin [42]. As a result, marine zooplankton dynamics represent an optimal candidate to inform marine policy and ecosystem management regarding environmental changes [43]. Despite the recognized critical role of zooplankton, challenges persist in assessing their response to climate change, and data accessibility remains a hurdle to gaining a comprehensive dynamic understanding.

Earlier investigations into changes in the planktonic community around the Portofino Promontory in the Ligurian Sea were conducted by Morabito et al. (2018) [44], and a more recent work has been carried out by Vassallo et al. (2021) [45] within the Long-Term Ecological Research (LTER Italia) framework. These studies, with a primary focus on copepods—small crustaceans that typically constitute 70% to 90% of mesozooplankton abundance—aimed to explore the dynamics of this vital ecological component. This study area presents an intricate circulation pattern influenced by wind direction and intensity, as well as the morphological features characterized by the interference of the promontory itself and the narrow continental shelf along the Ligurian coastline.

In this work, we employ prediction-based inferences using a machine learning and artificial intelligence (AI) methods to produce species distribution models (SDMs) [46,47] of the grey literature data from zooplanktonic Fabiano et al. (1988) [48] marine survey of 1985–1986. The digitalization of the historic grey literature not only meets the principles of FAIR data (findability, accessibility, interoperability and reusability) [49] but also makes vital information available to the public. In this context, this work allows us to shift back the zooplankton community information timeline to 20 years from first available data, presenting an important reference baseline for comparison before the trend of increasing temperature was first identified.

## 2. Materials and Methods

### 2.1. Study Area

The predominant large-scale characteristic of water dynamics in the Ligurian Sea is a cyclonic circulation that remains active throughout the year. This circulation is notably more vigorous during the winter months compared to summer ones, and it affects both deep and surface layers [50]. While climatic influences can lead to fluctuations in the intensity of fluxes, the overall pattern of this circulation is considered to be relatively constant [51]. This study focuses on the area surrounding the Portofino Promontory, a blunt headland with an abrupt, almost square shape (Figure 1) rising from the sea with very steep slopes. The Portofino Promontory has a very complex circulation due to both meteo-climatic (wind direction and intensity) and hydrodynamic (dominant circulation) forcing, as well as to the interference of the promontory itself together with the narrow continental shelf [52–55]. Current historical series show that the current off Sestri Levante in winter has a northwest direction (Ligurian Provençal current), consistent at all depths, while on the other side of the promontory, it has the opposite direction (southeast) with some variation in the vertical component. A recirculation or anticyclonic vortex near Camogli is triggered by the prevailing southeast current, as confirmed by historical records and numerical models [52]. Occasional current reversals off Camogli may occur due to local winds. Finally, off the coast of Bogliasco or off 15 km downstream of the promontory, the current again has a northwest direction and consistency at all depths, as confirmed using numerical models [52]. Along the western cape of the Promontory, the primary east-to-west stream can intensify and move away from the promontory under specific wind conditions. Meanwhile, winds originating from the south-southwest have the potential to strengthen coastal circulation within the Gulf of Tigullio. The Gulf receives inputs from several watercourses, resulting in a pronounced state of oligotrophy. This stream serves as a substantial source of freshwater, altering the physical, chemical and biological dynamics of the marine environment, impacting phytoplankton dynamics. In particular, the Entella River mouth, situated approximately 9 km to the east of the promontory, coupled with substantial anthropogenic influences (such as beach nourishment, excessive summer tourism, maritime activities and urban and industrial development), contributes significantly to sediment accumulation and heightened water turbidity, particularly on the eastern side of the promontory [56]. Since 1998, the promontory and its surroundings have been declared a Marine Protected Area (MPA), with the intent to preserve the coastal and marine ecosystem (the MPA's zonation is showed in Figure 1).

### 2.2. Data Elaboration and Modeling

In this study, the data employed were sourced from the grey literature, specifically digitized from the technical report authored by Fabiano et al. in 1988 [48]. The technical report, identified as No. 25 within the series of "Technical reports of the chair of Hydrobiology and Fisheries", was published by the Institute of Marine Environmental Sciences at the University of Genoa. Following Italian legislation, four copies are deposited at the Prefecture of Genoa and, therefore, readily available upon request (digitized copy is available in Supplementary Materials File S5). Original data collection took place on the Portofino Promontory over a year, from March 1985 to March 1986, with sampling occurring twice a month at fortnightly intervals. Sampling involved the use of paired "Bongo" nets, each featuring a 20 cm diameter mouth and 200 μm mesh, equipped with a General Oceanics Inc. model 2030 flow meter positioned at the net's center. The nets were towed to the surface for a duration of 20 min in two different locations nominally Chiavari A (44°15′1″ N–9°13′6″ E), located on the 200 m bathymetry, and Chiavari D (44°18′2″ N–9°18′2″ E), located on the 30 m bathymetry. Of the two collected samples, one was preserved in 4% formalin and utilized for both qualitative and quantitative analyses of zooplanktonic organisms. In the analysis, plankton samples were standardized to a constant volume of 500 mL. Following a homogenization procedure, subsamples of known volume were taken to taxonomically define no fewer than one hundred copepods, except in instances

from August, September and October, where the count concluded at fifty copepods due to their scarcity. The resulting copepod abundance tables, related to individual per cubic meter (Ind/m$^3$), from this study were digitalized and complemented with environmental data directly extracted from the technical report, when available (e.g., temperature, salinity, chlorophyll). The highest taxonomic resolution achieved in Fabiano et al. (1988) enables the differentiation of organisms in the analyzed samples up to the genus level. However, the absence of physical specimens stored in a permanent collection restricts further taxonomic investigations at the species level and presupposes the possibility of errors associated with misidentifications. Consequently, all analyses in this paper focused on modeling the distribution of copepods at the genus level, considering them to be assemblages of species. Additional proxies to increase the resolution and accuracy of the model were retrieved from the Copernicus Marine Service (https://marine.copernicus.eu/, accessed on 23 January 2024), NASA Earth Observations (https://neo.gsfc.nasa.gov/, accessed on 23 January 2024) and General Bathymetric Chart of the Oceans (https://www.gebco.net/, accessed on 23 January 2024) (See Supplementary Files S2–S4 for details). In our investigation, we employed the methodology proposed in Grillo et al. (2022) [57] with modifications. We utilized open-source software including QGIS [58], R [59] and Ocean Data View (ODV) [60] data exploration, visualization, mapping and modeling, incorporating basemaps where necessary. Geographic data were processed and displayed using the WGS84 EPSG:4326 projection system, facilitating latitude- and longitude-based analysis. To extract environmental descriptor values from the Copernicus Marine Service product layers, we overlaid presence data points onto the attribute table using the "extract multiple values in points" function within the GIS environment. The resulting table served as the basis for modeling copepod distribution using machine learning techniques, with subsequent representation in QGIS. For predictive modeling, we constructed a 1 km point lattice to generate a prediction grid, which was then evaluated using various machine learning algorithms. The environmental descriptors employed were sourced from open-access outlets and detailed in the online resource "Environmental descriptors" (see Appendix B). A Random Forest predictive distribution model was developed in R, drawing comparisons with similar studies such as those by Hardy [61] for Alaska, Meissner [62] for Iceland and Huettmann and Schmid [63] for Antarctica. For the analysis, a regression method was employed synergically with the 'rpart' (4.3.2 version) (https://cran.r-project.org/web/packages/rpart/index.html, accessed on 30 January 2024), 'randomForest' (4.3.2 version) (https://cran.r-project.org/web/packages/randomForest/index.html, accessed on 30 January 2024) and 'tidyverse' (4.3.2 version) (https://cran.r-project.org/web/packages/tidyverse/index.html, accessed on 30 January 2024) packages. We employed the original scripts developed by Guisan, Thuiller and Zimmermann (2017) [64] for predictive distribution analysis, tailoring them to accommodate the specific data matrix utilized in our study. Using the inverse distance weighting (IDW) tool within QGIS, we generated 24 predictive surfaces illustrating the relative abundance values (RA) of selected zooplankton genera across the entire study area, extending beyond the lattice point locations. This process facilitated the creation of a predictive surface grid. Subsequently, the predicted data were imported into ODV software to derive relative predicted abundance profiles for each copepod genus from the surface to the bottom. We focused on a specific section by selecting areas near the Portofino Promontory. By utilizing the gridded field function and DIVA gridding, we extrapolated values for the entire water column based on longitude. Initially, we crafted comprehensive distribution maps for copepods across the four distinct seasons using the complete data matrix. Our subsequent efforts were concentrated on the five most abundant genera of copepods, each representing a distinct trophic strategy (refer to Table 1 for details): *Acartia* spp., *Oithona* spp., *Centropages* spp., *Temora* spp. and *Coryaceus* spp.

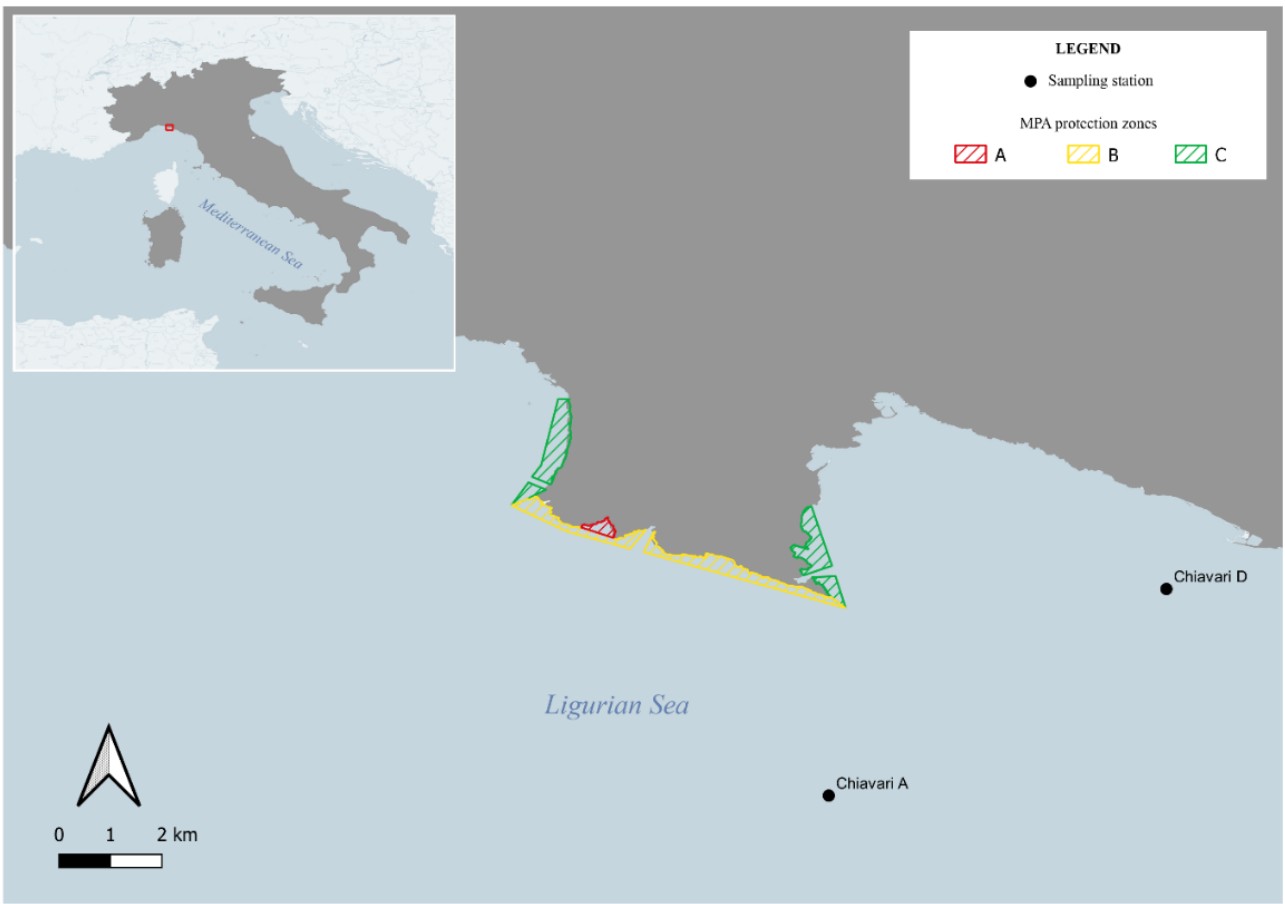

**Figure 1.** Study area. Black dots are original sampling stations from Fabiano et al. (1988) [48]. Portofino MPA marine protection zones are shown on the map (red = zone A, No-Take Zone; yellow = zone B, Sustainable Use Zone; green = zone C, Transition Zone).

**Table 1.** Genera analyzed in the present work. In this work, we analyzed the distribution of copepods at the genus level, considering them to be assemblages of species. AphiaID was from the World Register of Marine Species (https://www.marinespecies.org, accessed on 21 February 2024).

| Family | Genus | Worms Aphia ID | Trophic Guilds |
|---|---|---|---|
| Acartiidae | *Acartia* spp. Dana, 1846 | 104108 | Filter feeder |
| Oithonidae | *Oithona* spp. Baird, 1843 | 106485 | Ambush feeder |
| Centropagidae | *Centropages* spp. Krøyer, 1849 | 104081 | Suspension feeder |
| Temoridae | *Temora* spp. Baird, 1850 | 104241 | Filter feeder/Ambush feeder |
| Corycaeidae | *Corycaceus* spp. Dana, 1845 | 128634 | Predator |

We aimed to create detailed maps for individual seasons by focusing on these five genera. This was achieved by using the above algorithms and deriving the relative abundance (RA) for the lattice to identify the most suitable habitats for each copepod genus. We extracted the RA values for the analyzed points to assess how well the predictions aligned with the independent field data across the data. The accuracy of the selected models was assessed using the symmetric mean absolute percentage error (sMAPE), which measured the forecast accuracy based on percentage errors (see Table 2) [65–67]. sMAPE index was obtained from the 'Metrics' package (4.3.2 version) (https://cran.r-project.org/web/packages/Metrics/index.html, accessed on 30 January 2024).

**Table 2.** Genera list with sMAPE index accuracy values for each season. Red (very poor), orange (poor), yellow (good) and green (very good). sMAPE classes were modeled based on Chicco, Warrens and Jurman (2021).

| | sMAPE Index | | | | |
|---|---|---|---|---|---|
| **Season** | *Acartia* spp. | *Centropages* spp. | *Oithona* spp. | *Temora* spp. | *Coryceus* spp. |
| Autumn | 1.3698 | 1.0204 | 0.7357 | 1.0047 | 0.5111 |
| Winter | 1.0429 | 0.7985 | 0.4567 | 1.3779 | 0.8969 |
| Spring | 0.4105 | 0.9100 | 0.5558 | 1.8411 | 1.8448 |
| Summer | 0.9639 | 1.4982 | 1.0322 | 1.9648 | 0.8390 |

## 3. Results

We were able to compile, for the first time, a value-added data cube, explicit in time and space, consisting of copepod genera and environmental predictors to be used for model predictions of copepods for the Portofino Promontory area. Our field data consist of five genera with sample sizes, relative sMAPEs (Table 2) and RA values (Appendix A). Most of the sMAPEs obtained showed 70% higher accuracy in the season period investigated, which means that the analyzed models performed well with moderate accuracy.

The symmetric mean absolute percentage error (sMAPE) values present an insightful assessment of the predictive accuracy of copepod distribution models across different seasons and genera within the Portofino Promontory ecosystem. Each season reveals distinct patterns of predictive performance across the copepod genera, offering valuable insights into the underlying ecological dynamics.

In autumn, *Acartia* spp. demonstrates a relatively moderate sMAPE value of 1.3698, indicating a reasonable level of accuracy in predicting its abundance. Similarly, *Centropages* spp. and *Temora* spp. exhibit sMAPE values of 1.0204 and 1.0047, respectively, suggesting a comparable predictive performance. *Oithona* spp. follows suit with a relatively lower sMAPE of 0.7357, reflecting a closer alignment between the predicted and observed abundances. Notably, *Coryceus* spp. stands out with a notably low sMAPE value of 0.5111, indicating a high level of predictive accuracy for this species in autumn.

Moving into winter, the predictive performances of the models vary across genera. *Acartia* spp. and *Centropages* spp. display relatively lower sMAPE values of 1.0429 and 0.7985, respectively, suggesting a reasonable degree of accuracy in predicting their abundance. *Oithona* spp. shows a lower sMAPE value of 0.4567, indicating a closer alignment between the predicted and observed abundances compared to other genera. However, *Temora* spp. and *Coryceus* spp. exhibit higher sMAPE values of 1.3779 and 0.8969, respectively, suggesting greater discrepancies in the predictive models for these species during winter.

Spring unveils further nuances in predictive accuracy, with *Acartia* spp. displaying a notably low sMAPE value of 0.4105, indicating a high level of accuracy in predicting its abundance during this season. *Centropages* spp. follows suit with a moderate sMAPE value of 0.9100, reflecting a relatively consistent predictive performance across seasons. *Oithona* spp. and *Temora* spp., however, exhibit higher sMAPE values of 0.5558 and 1.8411, respectively, suggesting challenges in accurately predicting their abundance during spring. *Coryceus* spp. presents the highest sMAPE value of 1.8448, indicating significant discrepancies between the predicted and observed abundances.

Summer marks a period of increased complexity in copepod dynamics, as reflected in the sMAPE values across genera. *Acartia* spp. and *Oithona* spp. display moderate sMAPE values of 0.9639 and 1.0322, respectively, indicating relatively accurate predictions for these species. Conversely, *Centropages* spp. and *Temora* spp. exhibit higher sMAPE values of 1.4982 and 1.9648, respectively, suggesting challenges in accurately capturing their abundance during summer. *Coryceus* spp. stands out with a relatively lower sMAPE value of 0.8390, indicating a comparatively higher level of predictive accuracy for this species in summer.

The following are the two predictive maps that achieved the best sMAPE values in our work (sMAPE < 0.5).

Figure 2 displays the predictive map for *Acartia* spp., a filter feeder calanoid, during spring (sMAPE = 0.4105), along with the corresponding RA values. Regarding habitat suitability, the sMAPE index indicates high values (0.58) throughout the entire study area. The lowest RA values (300.18 Ind/m$^3$) are observed in the oceanic areas facing the Portofino Promontory, while the highest RA values (588.43 Ind/m$^3$) are found in the southeastern region of the promontory, close to the port and in the neritic zones.

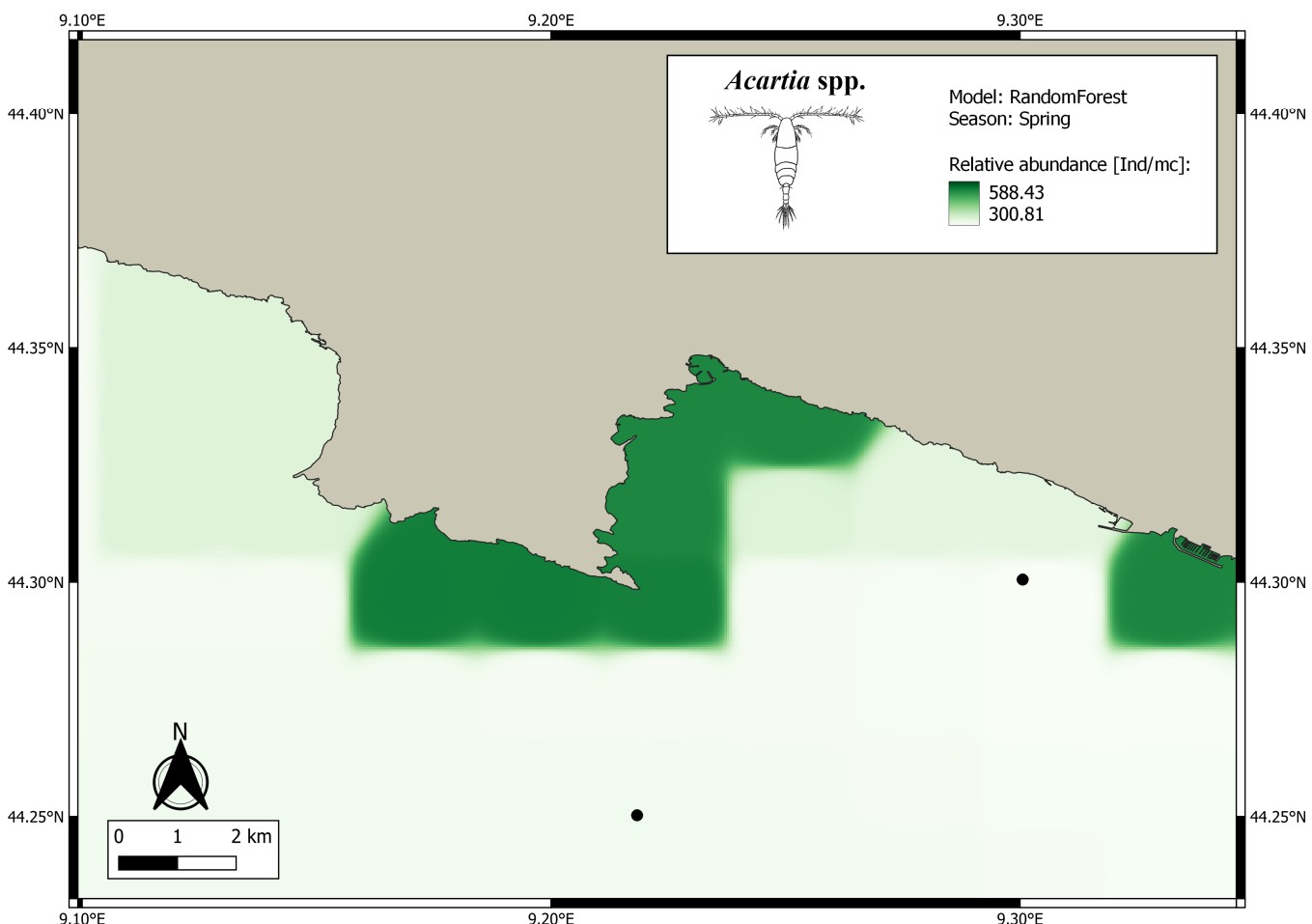

**Figure 2.** Predictive map for *Acartia* spp., a filter feeder calanoid, during spring (sMAPE = 0.4105), along with the corresponding RA values (expressed as Individuals/m$^3$). Black dots are original sampling stations from Fabiano et al. (1988) [48].

Figure 3 presents the predictive map for *Oithona* spp., an ambush feeder cyclopoid, during winter (sMAPE = 0.4567), accompanied by the corresponding RA values. Generally, *Oithona* spp. shows high RA values in all predicted areas. In detail, the highest RA values (80.06 Ind/m$^3$) are concentrated near the coastal zone, with medium-high values (57.95 Ind/m$^3$) observed in the pelagic zone. The lowest values (35.85 Ind/m$^3$) are sporadically found in the western zone and in proximity to the port within the study area.

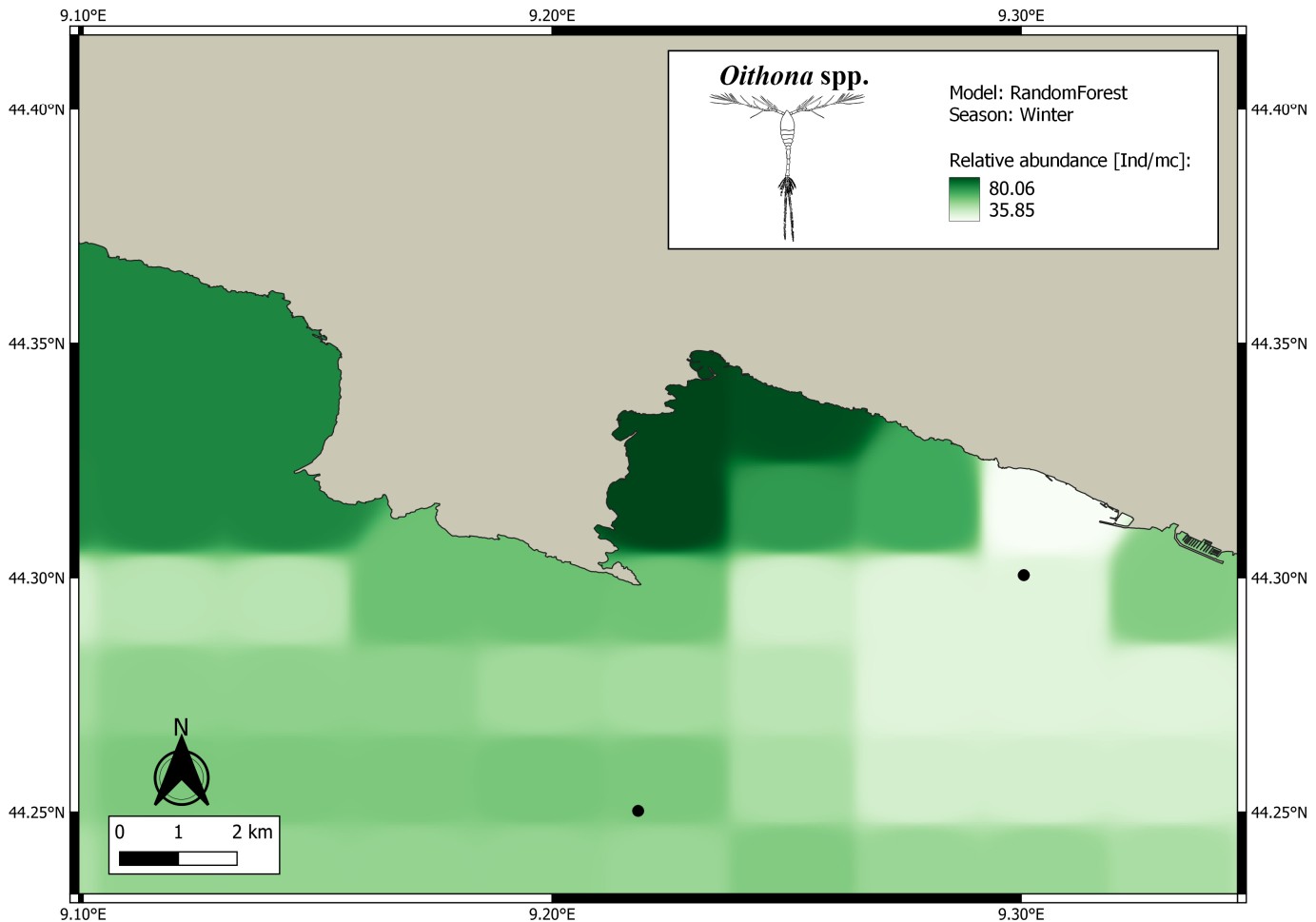

**Figure 3.** Predictive map for *Oithona* spp., an ambush feeder cyclopoid, during winter (sMAPE = 0.4567), accompanied by the corresponding RA values (expressed as Individuals//m$^3$). Black dots are the original sampling stations from Fabiano et al. (1988) [48].

All predictive distribution maps generated in this study, including those for other genera and various seasons, can be found in Supplementary File S1. Supplementary File S2 presents predictions regarding the abundance of copepods, from the sea surface to the seabed, in the areas close to the Portofino Promontory.

## 4. Discussion

The Mediterranean Sea, influenced by prominent climate patterns within the global climate system, emerges as one of the region's most susceptible to the impacts of climate change [68]. The Mediterranean basin plays pivotal role as a heat reservoir, source of moisture and support system for high marine biodiversity [25]. These environmental changes have a significant impact on marine communities, which may react with varying response times. Zooplanktonic communities exhibit sensitivity to environmental changes and react, for example, with biogeographical shifts [69], thus playing crucial roles as environmental indicators [38]. Copepods, among zooplankton, play a vital role in secondary production and grazing rates [70,71]. Understanding their potential distribution is, thus, crucial, given their ecological role as a trophic link between microzooplankton and secondary consumers [72]. However, very few works have been carried out on organisms that occupy the lowest trophic levels of marine food networks, despite their acknowledged paramount ecological role. To date, there is no work in the literature modeling the distribution of abundance or presence of any copepods in the Ligurian Sea. Most studies of species distribution models in Mediterranean or Ligurian Sea, in fact, were focused on secondary consumers,

top predators and alien species (e.g., Azzurro et al. 2013 [73], Giannoulaki et al. 2017 [74], Azzolin et al. 2020 [75] and Ranù et al. 2022 [76]).

In this work, we applied the methodology described in Grillo et al. (2022) [57] to the grey literature data from Fabiano et al. (1988) [48], leveraging pre-existing scripts in the scientific literature [64] and environmental data obtained from both the original technical report [48] and the Copernicus Marine Service to produce distribution maps. The inclusion of environmental descriptors sourced from open-access outlets enriched the predictive capacity of the models, which were evaluated by using a Random Forest machine learning algorithm approach (regression method). The focus on the five most abundant genera of copepods across different trophic guilds allowed for the creation of detailed distribution maps for individual seasons, shedding light on their habitat preferences and ecological dynamics. The outcomes derived from the developed models, based on a limited number of actual observation points, stand as a noteworthy achievement in unraveling the ecological intricacies of the Ligurian Sea. The capability to extrapolate meaningful data from a sparse set of observation points not only deepens our understanding of the ecosystem but also highlights the potential of these models as powerful tools for advancing marine ecology research and bolstering conservation initiatives.

The results obtained here by applying the above described approach represent a comprehensive and meticulous effort to model copepod distribution in the Portofino Promontory area, utilizing a combination of historical technical report and contemporary machine learning techniques. These "legacy data" enabled obtaining, for the first time, a 3D representation (i.e., with horizontal and vertical distributions) and SDMs for the most important genera of Ligurian Sea copepods in quantitative terms [45,77–79], i.e., *Acartia*, *Centropages*, *Oithona*, *Temora* and *Coryacaeus*, that have never been studied before in terms of habitat suitability.

The symmetric mean absolute percentage error (sMAPE) provides an assessment of the model accuracy of copepod distribution models in the Portofino Promontory ecosystem across different seasons. The observed variations in sMAPE values highlight the intricate interplay between environmental factors and copepod dynamics, shaping their abundance and distribution patterns. The relatively higher predictive accuracies of certain copepod genera, such as *Acartia* spp. and *Centropages* spp., during specific seasons suggest a marked seasonality in their ecological requirements and responses to environmental fluctuations. Conversely, the higher sMAPE values associated with genera like *Oithona* spp. and *Temora* spp. across multiple seasons indicate the challenges in accurately predicting their abundance, possibly due to the influence of additional ecological factors and chaotic dynamics, not considered here.

Our sMAPE results underscore the complexity of copepod dynamics within the Portofino Promontory ecosystem and highlight the need for the continued refinement and validation of predictive models to enhance our understanding of marine ecosystem dynamics across different seasons and copepod genera. Greater model accuracy can also be achieved by recognizing the importance of adding an additional level of refinement, working at the species level and incorporating species-specific responses and environmental variability into predictive models for copepod distribution. Future research efforts could focus on refining predictive models by incorporating additional environmental variables and fine-scale habitat features to enhance their accuracy and predictive capacity. Incorporating long-term monitoring data could provide valuable insights into the temporal trends and interannual variability in copepod abundance, aiding the development of more robust predictive models.

The trophic dynamics of Ligurian Sea zooplankton show a typical network of interconnected relationships, starting from the lower trophic levels represented by copepod genera, which are exploited by fishes. *Centropages* spp., in particular, which is the most abundant calanoid genera studied in temperate neritic zone [80] and present in all coastal stations analyzed here, represents an important element for modeling higher tropic levels. This genus has abundance hotspots in the functioning of seasons but generally, in neritic

provinces and near port. *Centropages* spp., represents up to approx. 6% of the stomach contents of prey [81]. This copepod genus has been found in stomach analyses conducted on small pelagic fish such as *Sardina* sp. and *Sardinella* sp. [82], as well as in mesopelagic fish such as *Cyclothone braueri* [83]. *Centropages* spp., as well as *Acartia* spp., *Oithona* spp., *Temora* spp. and *Coryacaeus* spp., are targeted by *Engraulis encrasiculus* (Linnaeus, 1758) [81], the European anchovy, a very abundant pelagic Engraulidae in Mediterranean fisheries that plays a very important role in marine food webs [84]. The prevalence of copepods in the ecosystem influences the abundance and distribution patterns of higher trophic level organisms, including migratory fish species and apex predators like tuna. This is reflected by the presence of the "Tonnarella di Camogli", a complex system of fixed nets designed to catch tuna fishes, which has been deployed along the western side of the Portofino Promontory (Golf of Paradise) since XVII Century [85].

This fishing structure obviously benefits from the increased abundance of copepods in the area due to a seasonal persisting vortex. This concentration of copepods, in fact, attracts larger fish, creating a hub of biological activity that extends throughout the food web. The vortex's influence on current patterns enhances the availability of food, ultimately impacting the movements and feeding behaviors of migratory fish species, whose movements also reflect the direction of the surface currents direction that arise in early spring [53]. The Tonnarella's exploitation of this dynamic environment capitalizes on the rich food resources provided by copepods, contributing to its significance as a strategic fishing location [86,87] that lasted for 400 years.

By modeling copepods' historical distributions, we can, thus, potentially discern how fluctuations in copepod populations influence the abundance and occurrence of higher-trophic-level organisms, including large predators like tuna, and obtain relevant data for sustainable fisheries management and ecosystem conservation efforts.

This process, however, is still hampered by our limited understanding of single-species ecologies, which are also greatly differentiated as a consequence of adaptation to different pelagic habitats.

Our data and metadata adhere to the FAIR (findability, accessibility, interoperability and reusability) principles [49]. The generation of three-dimensional predictive maps became feasible through the initial digitization of the "grey literature" data, which were subsequently repurposed to create 3D distributional maps. By adhering to the FAIR principle, distributional data extracted from these historical technical reports were effectively repurposed and recycled. This process significantly enhanced our comprehension of pelagic copepod biodiversity and contributed to refining the accuracy of our chosen models. These datasets will remain searchable, accessible and reusable to the fullest extent possible, providing a lasting glimpse into the structure of copepod diversity in the Ligurian Sea during the 1980s.

In conclusion, future advancements will focus on developing abundance distribution models using machine learning (ML) and artificial intelligence (AI) applications to predict the standing crop in the Ligurian Sea with greater quantitative precision. This effort aims to enhance monitoring capabilities for this area, which holds significance for mankind at a cultural, economic and ecological level. Overall, this approach not only advances our understanding of copepod ecology in the Mediterranean Sea but also serves as a template for integrating historical data with contemporary methodologies to elucidate marine ecosystem dynamics.

**Supplementary Materials:** The following supporting information can be downloaded via this link: https://www.mdpi.com/article/10.3390/d16030189/s1, File S1: Maps; File S2: Predicted data; File S3: Descriptor table; File S4: Raw species presence_absence data with environmental descriptors; File S5: Digitized copy of technical report. References [88–91] are cited in File S2.

**Author Contributions:** Conceptualization, A.G., M.B., S.S. and M.G.; methodology, A.G. and M.G.; formal analysis, A.G. and M.G.; resources, M.G.; data acquisition A.G., M.B. and M.G.; data curation, A.G., S.S. and M.G.; writing—original draft preparation, A.G. and M.G.; writing—review and editing,

A.G., M.B., S.S. and M.G.; funding acquisition, S.S. All authors have read and agreed to the published version of the manuscript.

**Funding:** This work was partially funded by the National Recovery and Resilience Plan (NRRP), Mission 4 Component 2 Investment 1.4—Call for tender No. 3138 of 16 December 2021, rectified by Decree No. 3175 of 18 December 2021 of the Italian Ministry of University and Research funded by the European Union—NextGenerationEU. Project code CN_00000033, Spoke 1, Concession Decree No. 1034 of 17 June 2022 adopted by the Italian Ministry of University and Research, Project title "National Biodiversity Future Center—NBFC" (G. Bavestrello).

**Institutional Review Board Statement:** Not applicable.

**Data Availability Statement:** All relevant data are presented within the manuscript and the provided Supplementary Materials.

**Acknowledgments:** We would like to express heartfelt thanks to Mauro Fabiano, the first author of the technical report used in this work, for authorizing the inclusion of the digitized copy of the original document in the Supplementary Materials. We are grateful to the two anonymous reviewers, whose comments and suggestions were of great help for improving the quality of this paper.

**Conflicts of Interest:** The authors declare no conflicts of interest. The funders had no role in the design of the study; the collection, analyses, or interpretation of data; the writing of the manuscript; and the decision to publish the results.

## Appendix A

The RAs (see Supplementary Materials File).

## Appendix B

Seasonal lattice grid (see Supplementary Materials File).

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
