# Peer review of "A Beacon in the Dark: Grey Literature Data Mining and Machine Learning Enlightening Historical Plankton Seasonality Dynamics in the Ligurian Sea"

_diversity, doi:10.3390/d16030189_

Round 1

Reviewer 1 Report

Comments and Suggestions for Authors

A Beacon in the Dark: Grey Literature Data Mining and Machine Learning Enlightening Historical Plankton Seasonality Dynamics in Ligurian Sea.

Alice Guzzi 1,2,3,*, Maria Balan 1,2, Stefano Schiaparelli 1,2,3 and Marco Grillo 3,4

The following manuscript discusses how historical data can be repurposed and reused by applying digitization. This work contributed mainly to the FAIR principle, which is nice. However, there are two main deficiencies in this research. Firstly, the copepods mentioned were not stored, and there is no possibility to check their identification. Secondly, the model was used not for species but for genera. These genera comprise numerous species, which can display different modes of feeding, resting, and reproduction. Therefore, the final result can be quite different. But since the authors themselves know it, believable that the model will be more precise for future modeling. 

Reviewer 2 Report

Comments and Suggestions for Authors

The manuscript is devoted to an interesting issue - the use of old data collected using classical methods to solve modern problems using modern modeling methods and assessing the accuracy of earlier studies. The main direction that interests the authors of the article is obtaining adequate data to assess ongoing climate changes. In this sense, the work seems relevant, quite new and in many respects promising. However, I have a number of comments that require responses/ changes before I can recommend the manuscript for publication.

The introduction should contain the specific aims of this study.

The manuscript does not have any clear conclusion that needs to be drawn, especially since the authors obtained an undoubtedly interesting result. In these conclusions it should be noted, for example, that based on the developed models for a small number of actual observation points, it was possible to estimate the total yield in the abundance of several genera throughout the Ligurian Sea, which is an important result and one of the goals of using such models in hydrobiology.

There is also a comment regarding the source of raw data. The publication place needs to be more precisely specified. Or, if this is an unpublished technical report, indicate where this data is stored and what the rights to use this data are, since the authors of these materials and the authors of the article are different people.
